# Effects of Aromatherapy on the Physical and Mental Health and Pressure of the Middle-Aged and Elderly in the Community

Mei-Hua Ke [1], Kun-Ta Hsieh [2,*] and Wen-Ying Hsieh [2]

1   Department of Health Beauty, Fooyin University, 151 Jinxue Rd., Daliao Dist., Kaohsiung City 83102, Taiwan; FT081@fy.edu.tw
2   Graduate School of Technological and Vocational Education, National Yunlin University of Science and Technology, 123 University Road, Section 3, Douliou 64002, Taiwan; hsiehwy@yuntech.edu.tw
*   Correspondence: dartribute@cyivs.cy.edu.tw; Tel.: +886-928778810

**Abstract:** The physical and mental health of an aging society has become a major issue, and stress reduction and the improvement of physical and mental health are important physical and mental health issues for middle-aged and elderly people. This research sought to explore the application of aromatherapy for the improvement of physical and mental health and stress levels, as well as other issues, that concern the elderly in the community. The research was based on intentional sampling. A pre- and post-test design with unequal groups was employed. The experimental treatments were divided into five groups: Group A (compound essential oil massage plus sniffing), Group B (compound essential oil massage), Group C (pure base oil massage), Group D (compound essential oil sniffing), and control Group E (without any aromatherapy intervention). To explore the effects of aromatherapy on physical and mental health and stress relief among the elderly in the community, the self-completed Mental and Physical Health Scale for the Elderly and the Stress Index Measurement Scale were used to collect data. The obtained data were analyzed using descriptive statistics and by paired sample *t*-test. It was concluded that aromatherapy can improve the physical and mental health of the elderly in the community and can significantly reduce stress. The experimental results on aromatherapy in this study can provide a basis for home application to help the elderly in the community. They also provide a foundation for the organization of health promotion courses for the elderly and other practical applications in social welfare group planning.

**Keywords:** aromatherapy; middle-aged and elderly; physical and mental health; stress

## 1. Introduction

### 1.1. Research Background and Motivation

In an aging society, population aging is accompanied by many leisure, economic, care, medical, psychological and social problems. The health status of the elderly is closely related to the development of an aging society [1] (Chen Hongshun, 2010). The middle-aged and the elderly have a high demand for relaxation, stress relief and healthy living. Therefore, they will seek ways to help them achieve harmony between body, mind and spirit. Among these, aromatherapy is one of the most popular alternative medical treatments. Having emerged in modern times, aromatherapy has been shown by numerous studies to be able to improve people's emotional condition, sleep, and physical and mental health problems caused by stress. The method is simple and the effect is good. Widely accepted by the public [2–8], aromatherapy is viewed as an auxiliary and natural therapy. Users can simply use essential oils extracted from plants as a means of preventing health problems and regulating their physical and mental health condition.

It has been found that aromatherapy has effects on people's physical and mental health and stress. However, there are very few studies that have focused on how the use of compound essential oils can help with the physical and mental health and stress of the

elderly in the community. With a view to learning more about the effects of aromatherapy for this particular demographic group, this study focuses on middle-aged and elderly people in the community as the research population, and considers the practical application of aromatherapy to improve the physical and mental health and reduce stress of middle-aged and elderly people in the community. In addition to affecting quality of life, stress is also closely related to human physical and mental health. Through the process of good stress management and positive adaptation, the elderly can increase their self-efficacy and happiness [9]. With the fast-paced and busy life of modern society, people experience pressure from all directions. When the pressure exceeds the psychological capacity to cope, there will be maladaptation. In severe cases, it may even cause mental illness (Department of Psychiatry, Hospital Affiliated to National Taiwan University School of Medicine, 2020). Ref. [10] pointed out in the article *Stress and Modern Life* that, over five million years of human evolutionary history, the modern way of life now involves fewer activities than at any other times. The way people live and work has never been more stressful, while doctors suspect that as many as 70% of diseases are related to high blood pressure. Pressure-related headaches and migraines are the most common physical symptoms experienced by people who are stressed or anxious, though there are many other stress symptoms [10].

The pace of modern society is fast. Most people are busy with work and are physically and mentally tired. Work pressure, family pressure and social pressure often cause people to suffer lack of sleep, which can, in turn, lead to various physical and mental health problems, affecting the quality of life and physical and mental health. These problems are particularly prevalent in middle-aged and elderly people. Elderly people have many stress and sleep problems [11]. The health of middle-aged and elderly people is a most important concern and requires attention in the future. Therefore, how to ensure that middle-aged and elderly people live a healthy and happy life is a most important issue for our aging society. To achieve successful aging, older adults need to maintain positive emotions to enhance mental health. However, in recent studies, late-stage depression and anxiety have been found to be prevalent and have become major problems in society [11]. Therefore, how to reduce stress and relieve physical and mental health problems is the most important physical and mental health care issue for middle-aged and elderly people.

Aromatherapy is a way to relieve physical and mental stress by improving physical and psychological states using essential oils extracted from aromatic plants through inhalation, massage, bathing, incense and other conditioning methods. It is a natural therapy for disease prevention. At present, many elderly care institutions in Taiwan use aromatherapy extensively to help residents and cancer patients relax and open their minds, and to improve the physical and mental condition of disabled people [12]. Our country has become an aging society. Aging is accompanied by an increase in chronic diseases. As a result, more and more middle-aged and older Chinese people are paying attention to the maintenance of their physical and mental health and are actively seeking health care. Aromatherapy is becoming more and more important. It is one of the health care methods used by the elderly in modern society. It can be integrated into home health care to promote physical and mental health and to improve quality of life [13,14].

*1.2. Research Purpose*

The purpose of this study was to investigate the effects of aromatherapy on physical and mental health and stress relief among the elderly in the community. This was achieved by evaluating the differences between a treated experimental group and a control group which did not experience an aromatherapy intervention, before and after the intervention. The research findings provide a basis for the use of aromatherapy at home by the elderly in the community to improve their physical and mental health and reduce stress.

## 2. Literature Review

### 2.1. Aromatherapy

Aromatherapy is defined in the revised version of the *Mandarin Dictionary* by the Ministry of Education as an unorthodox treatment method that uses the aroma of essential oils extracted from various plants to refresh people's spirits, nourish the body and heal diseases. The definition found in the Oxford dictionary is "the use of essential oils extracted from aromatic plants for therapeutic and cosmetic purposes." [5] also pointed out in his research that aromatherapy involved the use of essential oils extracted from plants through massage, bathing, inhalation, and tea drinking to improve the body's self-defense ability and to prevent or treat diseases. Ref. [15] stated in the book *Aromatherapy* that the term aromatherapy comes from the essential oils that are extracted from plants and have a natural aroma. The effect is to achieve a natural and safe curative effect, and provides comprehensive therapy. Ref. [16] also proposed in the book *Aromatherapy and Body Care* that aromatherapy can be interpreted as "a therapy that uses the aroma of natural plants to heal physical and mental diseases". In addition, Ref. [17] stated in *Aromatherapy-Basic Mechanisms and Evidence Based Clinical Use* that aromatherapy is a practice that utilizes the health properties of therapeutic plants and essential oils. Ref. [18] pointed out that aromatherapy is defined as the absorption of aromatic essential oils through breathing or massage to relieve emotions, relax the mood, or improve physical diseases, etc. Ref. [19] also stated in the *Cochrane Collaboration* that aromatherapy is the extraction of pure essential oils from aromatic plants to help solve health problems and to improve quality of life; they stated that it is a branch of herbal medicine involving the extraction of pure essential oils from the roots, stems and leaves of plants [14,20–22].

### 2.1.1. Aromatherapy Conditioning

Aromatherapy is the therapeutic use of essential oils, primarily through inhalation, but also through the skin, to treat, reduce or prevent disease, infection and discomfort [17]. Ref. [23] in Weiner's *Pain Management—A Practical Guide for Clinicians—Aromatherapy for Pain Relief* stated that essential oils can be diluted in a bathtub or diffused into the air through a burner, massaged with a small amount of oil, or simply inhaled from a burner, and that this may be much safer than the use of oral complex oils. Aromatherapists use various means of extracting aromatic essential oils from plants. They aim to use pure essential oils to keep their composition as constant as possible so that the oils have predictable properties. Aromatherapy is a method of physical and mental relief that is accepted by the public involving the use of essential oils extracted from natural plants, supplemented by massage, inhalation, bathing, and aromatherapy [24]. The most used aroma conditioning methods are the massage method and the inhalation method. Common inhalation methods are: (1) the inhalation method; (2) the aromatherapy method; and (3) the spray method. Aromatherapy uses essential oils for conditioning purposes and uses odor therapy. The aroma molecules of essential oils are transmitted to the brain and affect the emotions. Through the emotional response of the brain, physical and psychological problems can be improved. The main route for absorption is through the nose or through the skin. Absorption of the oils helps regulate physical and mental conditions and prevent diseases.

### 2.1.2. Aromatherapy for the Elderly

Ref. [25] pointed out that aromatherapy can be quite helpful to silver-haired people who seek to relax, relieve stress and live a healthy life. The elderly often use natural preventive medicine therapies, and subscribe to the view that "prevention is better than cure". Silver-haired people emphasize the harmony of body, mind and spirit in search of anti-aging and eternal youthful vitality. They are also very interested in spiritual and emotional healing courses. Ref. [26] pointed that there are many modern studies on aromatherapy for the elderly. In one study, it was found that, if used properly, aromatherapy can improve depression and cardiovascular diseases, which can not only reduce medical costs, but also improve the quality of life of silver-haired people. Ref. [12] pointed out in

her study *The Application of Aromatherapy in Elderly Care Institutions* that aromatherapy can relieve stress and depression, lower blood pressure and improve cardiovascular function. At present, many elderly care institutions in Taiwan make extensive use of aromatherapy to help the elderly relax and improve the physical and mental conditions of the disabled through aromatherapy.

Our country has entered an aging society. As people grow older, their bodies age, and chronic diseases increase. Therefore, Chinese people above middle and old age should pay more attention to the maintenance of physical and mental health, and actively seek ways to maintain health care. Aromatherapy is suitable for middle-aged and elderly people. The health care methods used by the elderly can be integrated into home health care to promote physical and mental health and improve the quality of life [14].

### 2.1.3. Essential Oils

Essential oils are the essences and aromatic substances extracted by physical means, but only trace units can be extracted from specific plants. Plant essential oils are organic liquids extracted from plant raw materials; because of their volatility, they are released to the air. Essential oils can emit fragrance when fresh [15]. Plant essential oils can be divided into two types: (1) the essence extracted from plants, i.e., essential oils, which are extracted from different parts of plants; (2) the base oil, including sweet almond, grape seed, soybean, wheat germ, jojoba oil, etc. [27].

### 2.1.4. Single Essential Oils and Base Oils Used in This Study

In this study, a single essential oil with relaxation, soothing and sedative properties was selected; lavender, bergamot, marjoram and sweet almond base oils were blended into a 5% concentration of compound essential oils to explore whether the use of aromatherapy affects the physical and mental health and stress of the elderly in the community.

(1) Lavender

Lavender is a multi-purpose plant essential oil, whose essential oil properties can be anti-inflammatory, pain relieving, anti-infective, sedative and soothing. It has an excellent soothing effect on the nervous system, can balance and regulate the central nervous system, and can effectively relieve symptoms such as nervous insomnia, headache, migraine, depression, anxiety, and bipolar disorder caused by nervous system imbalance. It also has soothing, sedative and calming effects on the circulatory system.

(2) Bergamot essential oil

Bergamot, a member of the Rutaceae (or Citrus) family, is mainly produced in Italy and the Ivory Coast. Its mild and uplifting aroma is often used to relieve stress and anxiety, while the role of bergamot essential oil itself is often used for various types of infections. The essential oil properties include a fragrant and fresh smell, and it has the effect of calming, soothing and boosting the spirit. It can also help reduce inflammation of the skin and sterilize and heal wounds. It can regulate the autonomic nervous system, and relieve symptoms caused by depression, anxiety and stress.

(3) Marjoram essential oil

Marjoram, also known as fragrant mint, belongs to the Origanum genus of the Labiatae family. Marjoram essential oil has a calming and sedative effect, helping to relieve anxiety and tension; it can also relieve headaches, including parietal headache.

(4) Sweet Almond Oil

Sweet almond oil is obtained from sweet almonds. It is extracted by cold pressing. The oil color is pale white to yellow. Sweet almond oil contains a lot of oil components. It is the basic oil for diluting single essential oils. It can make the skin smooth and soft and can reduce irritation. It has good affinity, is a smooth oil, and contains mineral protein and various vitamins. It can promote cell renewal, relieve itching, reduce skin dryness and inflammation. Therefore, it has become the most common base oil in massage oil and aromatherapy.

To verify whether the combination of essential oils and aromatherapy massage can improve physical and mental health and stress in middle-aged and elderly participants, three soothing and anti-stress essential oils, lavender, bergamot, and marjoram were selected for research. Through the combination of essential oils and aromatherapy massage on the head, shoulders and neck, the research participants reported experiencing a relaxing and soothing effect.

### 2.2. *Middle-Aged and Elderly People, Physical and Mental Health, Stress*

2.2.1. Age Definition of Middle-Aged and Elderly People

A definition of middle-aged and elderly people in our country is provided in *Employment Promotion Law for the Middle-aged and the Elderly and the Elderly* (2019):

(1)    Middle-aged and elderly refers to persons between the ages of 45 and 65.

(2)    Elderly refers to persons over 65 years old.

The "elderly person" according to our country's *Elderly Welfare Law* (2015) refers to those who are over 65 years old, while the *United Nations Population Statistical Yearbook* identifies those over the age of 65 as elderly [28].

According to relevant laws and regulations, in our country, people between the ages of 45 and 65 are defined as middle-aged and elderly, and those over 65 are defined as seniors [14].

2.2.2. Physical and Mental Health

As early as 1948, the World Health Organization (WHO) defined health as "a state of complete physical, psychological and social well-being, not merely the absence of disease or infirmity". Physical and mental health refers to the condition that an individual has no disease in their body organs, feels happy and satisfied psychologically, gets along well with others in society, and feels comfortable and unburdened both physically and psychologically [29].

This study aims to understand the cognition of the elderly in the community and their physical and mental health status and social participation. Physical and mental health is divided into four levels: physical health, mental health, social participation, and overall physical and mental status.

(1)    Good health

Physical health includes two aspects: First, the main organs function normally, the body shape is well developed, the body shape is uniform, the various systems of the human body have good physiological functions, and the physical activity and labor capacity are relatively strong; Second, resistance to disease, and ability to adapt to environmental changes, various physiological stimuli and the effects of pathogenic factors on the body [30]. As people grow older, more and more problems arise in terms of physical function, with daily activities and physical health causing varying degrees of distress to the elderly in the community.

(2)    Mental health

The World Health Organization defines mental health as "a state of health and well-being in which a person can achieve his abilities, cope with the stresses of daily life, be productive at work, and contribute to all that he has made" [31]. In Young's *Help Yourself Towards Mental Health* (2018), mental health is defined as having the ability to self-realize, which implies activity, a high degree of authenticity, and a capacity to be inward-looking and self-reflective. Mental health can be conceptualized as a state of well-being in which individuals can realize their abilities, can cope with the normal stresses of life, can perform productive work and can contribute to their community.

(3)    Social participation

Social participation can be defined as an individual's voluntary participation in community courses, clubs, activities, and their participation in social affairs and interaction

with the public, enriching life, and achieving self-satisfaction [32]. Participating in community activity courses can increase interaction between people, enhance emotions, promote interpersonal relationships and provide a sense of identity, increase self-confidence, and make people spiritually satisfied.

(4)　Overall physical and mental condition

The overall physical and mental state refers to an individual's satisfaction and happiness in terms of physical functioning and activity, the degree of satisfaction and happiness with their psychological, emotional, and living conditions, involvement in social relations and activities and ability to get on with others, and feelings of peace, comfort and satisfaction.

### 2.2.3. Pressure

The term stress was first extended from physics in the 1920s, mainly to describe an external force that can cause an object to deform. It was developed by the physiologist Walter Cannon in 1926 to characterize negative effects on the ability to maintain a constant internal physiological state (homeostasis) [33,34]. The so-called internal constant is the optimal range required to maintain the body's operation; the term stressor refers to the factors that generate a stress response, which may be internal or external.

Stressors can generally be divided into two categories: physiological and psychosocial stressors. Physiological stressors refer to factors that directly challenge the body's constants, such as hunger, cold, or illness, while the latter refers to people's cognitive judgments that external factors may be harmful to themselves. Some stressors may initially come from physical discomfort and turn into psychological stress, for example, emotional fear relating to understanding of disease, fear of the disease worsening, or excessive worry about the cause. Sexual stressors can lead to physical discomfort, such as stomach pain caused by excessive tension [33]. Stress, which implies tension, refers to a state in which individuals often feel oppressed [35]. Stress refers to a certain reaction of an individual when faced with a stressful event and the resulting comprehensive psychosomatic state [36,37].

### 2.2.4. Research on Aromatherapy and Physical and Mental Health and Stress

With the rapid changes and development of society, the activities of human beings have changed. In a busy life, people are faced with various pressures every day. When the pressure cannot be relieved and the load is excessive, this can affect physical and mental health [35,38]. Ref. [39] researched the use of 3D VR immersive reality, through an interactive course, to improve the promotion of aromatherapy education, and to verify the impact of aromatherapy on the physical and mental health of the elderly using scientific methods. The results showed that the 3D VR aromatherapy course was able to effectively promote the health of the elderly through the combination of reality and virtual reality. Ref. [40] researched the effect of an aromatherapy acupoint massage course on the perceived health of silver-haired people. The results showed that: (1) The aromatherapy acupoint massage course was able to improve the perceived health of silver-haired people; (2) With the intervention of aromatherapy and acupoint massage courses, the mental health status of silver-haired people in community care was significantly improved; (3) Through the intervention of aromatherapy and acupoint massage courses, the perceived health status of silver-haired people in community care was significantly improved; (4) Through the intervention of aromatherapy and acupoint massage courses, the perceived social health status of silver-haired people in community care was significantly improved.

Aromatherapy has been applied to relieve psychological stress. Aromatic substances (including essential oils, plant volatile oil extracts, etc.) can soothe emotions, invigorate spirit, eliminate depression, and enhance self-confidence, and can reduce patients' tension and negativity [41]. Aromatic essential oils have a fragrant smell. The chemical molecules of plants enter the human brain through the olfactory system with the aroma, thereby regulating the nervous system, helping the normal secretion of human hormones, and reducing physical discomfort. These aromas can also have a strong relaxation effect, regulating

emotions. Therefore, more and more people are willing to accept this technique to relieve various pressures [42]. Ref. [9] explored the effect of aromatherapy on the stress of middle-aged and elderly people. They found that the experimental group receiving compound essential oil massage showed a significant improvement in stress, while the control group showed no significant improvement in stress. It has been shown that aromatherapy can effectively relieve and improve the stress of middle-aged and elderly people. Ref. [43] study pointed out that aromatherapy has a wide range of applications. In addition to palliative care, it is also very helpful for the relief of symptoms such as stress, insomnia and anxiety. It has also developed into a new leisure and health culture for Chinese people in Taiwan. Research has shown that most people have physical and mental stress symptoms from various stress sources, and the greater the level of physical and mental stress, the greater the awareness of the degree of stress relief provided by of aromatherapy.

### 2.2.5. Research on the Physical and Mental Health and Stress of Middle-Aged and Elderly People

Having good physical health is the primary condition for successful aging, and is also an objective indicator of quality of life. When evaluating physical health, it should be evaluated from both subjective and objective aspects [44]. Ref. [45] explored the impact of middle-aged and elderly learners' participation in senior learning on their physical and mental health. The results showed that the physical and mental health performance of senior learners was not affected by gender, age or education level, but was affected by health status and economic status. Nor was the physical and mental health performance of senior learners affected by different course types. Rather, the more courses pursued and the longer the elderly participated in learning activities, the better they performed in terms of physical and mental health.

Ref. [46] pointed out that aging, disability, and mental or physical diseases are factors that affect the long-term use of sleeping medication by middle-aged and elderly people. Studies show that middle-aged and elderly people are worried about a range of problems, including health problems, the inconvenience caused by disease, and how they are getting along with their family members. All of these might increase the possibility of using sleeping pills. Ref. [47] studied the factors that affect the work stress of middle-aged and elderly workers. It was found that 40- to 60-year-olds had the highest work-related stress, 40– to 50-year-olds had the highest work-related stress response, and middle-aged and elderly workers had higher work-related stress than non-middle-aged and elderly workers. The stress response was higher than that of the middle-aged and the elderly. Middle-aged and elderly people in non-management positions had higher work stress and higher work stress response.

### 2.3. The Use of Aromatherapy in Combination Therapy

Combination therapy (polytherapy), is a therapy that uses more than one medical treatment. Generally, this term refers to the use of multiple therapies to treat a condition. Generally, combination therapy uses drugs (although some non-medical means are sometimes used to treat depression, such as drugs and psychological counseling to treat depression) [48]. At present, combination therapy mostly uses a combination of traditional Chinese and western medicine. In a study by [49], concerning the clinical application of Chinese and Western medicine to cancer patients, it was pointed out that the comprehensive treatment of Chinese and Western medicine includes conventional treatment, function rehabilitation, psychiatric and psychological support, traditional Chinese medicine and acupuncture, and tranquility care. The purpose is to restore the patient's physical health and spirits, so that the patient can return to social life. Combination therapy is also seen as a modern approach to the comprehensive rehabilitation of older adults, affecting stress, and cognitive and motor function. Aromatherapy is regarded as an adjunctive therapy that can help middle-aged and elderly people to improve their physical and mental health and

stress. Through experimental research, aromatherapy has been shown to be an excellent choice for a comprehensive rehabilitation approach.

## 3. Material and Methods

### 3.1. Research Design

This study took middle-aged and elderly people over 55 years old in Kaohsiung City as the research population and aimed to explore the effect of aromatherapy on the physical and mental health and stress of middle-aged and elderly people in the community. As well as making a comparison between experimental groups treated with aromatherapy and a control group without any aromatherapy, our research also sought to determine the differences between subjects before and after the intervention in each group.

At the beginning of the research, we applied for ethical review for human research in Chenggong University. The experimental process was only started after the review was completed and approval of the committee granted. Before recruiting experimental participants, we required the consent of community gatekeepers. When inviting community elders to participate in the study, we undertook due diligence by first informing them of the experimental methods and procedures before agreement and the signing of an informed consent form.

The research experiment involved the elderly and the elderly in the community as participants. The study involved a quasi-experimental design with intentional sampling. The subjects were divided into four experimental groups and one control group. In the experiment, the groups were as follows: Group A were treated with compound essential oil massage plus inhalation; Group B were treated with compound essential oil massage; Group C were treated with pure base oil massage; Group D were treated with compound essential oil inhalation. Participants in the control group, Group E, just went about their daily activities without any aromatherapy intervention. The experimental groups received compound essential oil head, shoulder and neck massage and inhalation for a total of eight weeks, once a week, with massage for 30 min on each occasion, inhalation for 15 min on each occasion, with massage in the first week, before inhalation and in the eighth week. After sniffing, subjects had to fill in the physical and mental health scale and the stress index measurement table. Participants in the control group also had to fill in the physical and mental health scale and the stress index measurement table in the first and eighth weeks of the experiment (Table 1). After the experiment, the physical and mental health and stress scores were calculated to evaluate the difference in the improvement of physical and mental health and stress of each subject receiving compound essential oil massage and compound essential oil inhalation.

**Table 1.** Physical and mental health scale for middle-aged and elderly people.

| Scale Facets | Item |
|---|---|
| Physical health | 1. Are you satisfied with your current physical health? <br> 2. Are you satisfied with your current sleep status? <br> 3. Are you satisfied with your current daily activities? <br> 4. Are you satisfied with your current level of physical activity (moving around, exercising, climbing stairs, etc.)? |
| Mental health | 1. Are you satisfied with your current attitude towards life? <br> 2. Are you satisfied with your current emotional state? <br> 3. Are you satisfied with your current state of mental activity in all aspects? |
| Social participation | 1. Are you satisfied with participating in community class activities? <br> 2. Are you satisfied with the relationships you have built by participating in community class activities? <br> 3. Are you satisfied with your current relationship? |
| Overall physical and mental status | 1. Overall, are you satisfied with your current physical health? <br> 2. Overall, are you satisfied with your current mental health situation? <br> 3. Overall, are you satisfied with your current social participation situation? |
| Total | 13 |

*3.2. Research Subjects*

This study explored the effect of aromatherapy on the physical and mental health and stress of the elderly in the community. Elderly people in the community were invited to fill in the physical and mental health and stress scale for the purpose of the experimental research. In order to ensure the accuracy of the questionnaire, the sampling was intentional in the sense that only elderly people were invited who were able to move freely in the community care bases and senior learning centers. Elderly people resident in nursing centers, long-term care institutions, or with limited mobility were excluded from the sample. The participants in the study were recruited from Kaohsiung City, were aged over 55 years old, and could take care of themselves. They were divided into four experimental groups and one control group, with 15 people in each group, totaling 75 people.

*3.3. Research Tools*

3.3.1. Physical and Mental Health Scale for Middle and Old Age

This research used the self-completed questionnaire, the Physical and Physical Health Scale for Middle-aged and Elderly. There were a total of 13 questions, divided into four parts, except for basic information on the respondents. Questions were rated according to a five-point Likert scale, with response levels of "Very satisfied", "Satisfied", "Normal", "Dissatisfied", and "Very dissatisfied". The respondents selected the most suitable answer according to the degree of compliance. The options were scored as 5 points, 4 points, 3 points, 2 points and 1 point, the higher the score, the higher the degree of satisfaction, and vice versa, the lower the score, the lower the satisfaction.

In terms of reliability analysis, the Cronbach's alpha coefficients of the four subscales were as follows: Physical health subscale 0.714, Mental health subscale 0.938, Social participation subscale 0.922, Overall physical and mental status subscale 0.780. The reliability coefficients for the four subscales were all above 0.70, indicating high internal consistency and good reliability. The alpha coefficient for the total scale was 0.890, which indicated that the reliability of the scale was quite good.

3.3.2. Pressure Gauge

The stress scale for the elderly used in this study was the stress index measurement table of the National Health Administration of the Ministry of Health and Welfare of the People's Republic of Taiwan. This scale is published by The National Health Administration of the Ministry of Health and Welfare on the website for the public to use to understand their own stress status.

There are 12 questions in total in the scale, including: "Have you been nervous recently and felt that you can't finish your work?", "Have you been sleeping poorly recently, with frequent insomnia or poor sleep quality?", "Do you often feel depressed, anxious, or irritable?", "Have you been forgetting things and becoming very forgetful recently?", "Have you been losing your appetite?" or "Has your appetite been particularly good?", "Have you been sick more than once in the past six months?", "Have you been feeling very tired recently and have you been sleeping on holidays?", "Have you often felt headaches and backaches recently?", "Have you often had disagreement with others?", "Have you been having trouble concentrating recently?", "Have you been feeling uncertain about the future? Or fearful?", "Has anyone said you don't look well recently?".

Answering "yes" or "no" by ticking the boxes, the respondent is required to select items that match their physical and mental state according to their own interpretation. Higher scores indicate higher levels of stress, whereas lower scores indicate lower levels of stress. If the respondent has three "yesses", it means that the respondent's stress index is still within the acceptable range. If the respondent has four to five "yesses", it means that the respondent is experiencing a high level of stress, and that, being barely able to manage, the respondent should study stress management seriously, and chat with teachers and friends. If the respondent has six to eight "yesses", it means the respondent is under a great deal of stress, and should see a mental health professional and receive systematic psychotherapy.

If the respondent has nine or more "yesses", it means that the stress is so serious that the respondent should see a psychiatrist and use drug therapy and psychotherapy as prescribed by the doctor to help their life return to normal as soon as possible (National Health Administration Health 99 website, 2019) [50].

### 3.3.3. Essential Oils

The essential oils used in the experiment were products of the general agent of Shangxiong Biotechnology Industrial Co., Ltd., Tainan City, Taiwan which had passed the ISO 9001-2000 standard certification for production of pharmaceutical-grade high-quality essential oils. The essential oils used in the experiment were lavender essential oil, bergamot essential oil, marjoram essential oil and sweet almond oil. We added 4 drops of lavender essential oil, 4 drops of bergamot essential oil, and 2 drops of marjoram essential oil to 10 mL of sweet almond oil as a compound essential oil blending design and prepared a compound essential oil formula with a concentration of 5%.

### 3.3.4. Data Analysis

In this study, using the data obtained from the scales, the SPSS 22.0 Chinese version software package (Taiwan) was used for statistical analysis, and the data were analyzed using descriptive statistics and paired sample *t*-test statistical methods.

(1)  Descriptive statistics

Descriptive statistics is a general term in statistics used to describe or summarize the basic situation for observations made. The valid data obtained by use of the research questionnaire were presented in terms of descriptive statistics including the mean and standard deviation to illustrate the status of each variable and each dimension. We also used the mean and standard deviation to summarize the "physical and mental health" and "stress" scores of the elderly in the community.

(2)  Paired-sample *t*-test

A paired sample *t*-test for dependent samples is commonly used in a repeated measures design with dependent samples; that is, the same sample is measured twice before and after. A paired sample compares the mean difference between two sets of dependent samples. In this study, a dependent samples *t*-test was used to test whether the pre- and post-test scores of "physical and mental health" and "stress" of the experimental groups, and the pre- and post-test scores of "physical and mental health" and "stress" of the control group, reached a significant level.

## 4. Results and Discussion

### *4.1. Results*

#### 4.1.1. Difference Analysis of the Overall Pre- and Post-test of "Physical and Mental Health" among the Elderly in the Community before and after the Aromatherapy Intervention

From Tables 2 and 3, the difference analysis results for the pre- and post-test of the "physical and mental health" of the elderly in the community were as follows: there were significant differences in "physical and mental health" between the pre-test and post-test in Group A, Group B, Group C, and Group D, and only Group E did not show a significant difference.

#### 4.1.2. Before and after the Aromatherapy Experiment: Difference Analysis of the Pre- and Post-Test of "Stress" among the Elderly in the Community

From Tables 4 and 5, the difference analysis results of the pre- and post-test of the "stress" construct of the elderly in the same group of communities were as follows: Group A, Group B, and Group D showed significant differences in "pressure" before and after the intervention, but there was no significant difference observed for Groups C and E.

**Table 2.** Summary table of the overall pre- and post-test mean and standard deviation of "physical and mental health" for middle-aged and elderly people in the community.

| Group | Test | Number | Average | Standard Deviation | Standard Error |
|---|---|---|---|---|---|
| Group A | Pre-test | 15 | 3.678 | 0.409 | 0.106 |
| | post test | 15 | 4.465 | 0.395 | 0.102 |
| Group B | Pre-test | 15 | 3.846 | 0.185 | 0.048 |
| | post test | 15 | 4.100 | 0.253 | 0.065 |
| Group C | Pre-test | 15 | 3.760 | 0.410 | 0.106 |
| | post test | 15 | 4.056 | 0.435 | 0.112 |
| Group D | Pre-test | 15 | 3.653 | 0.461 | 0.119 |
| | post test | 15 | 4.011 | 0.278 | 0.072 |
| Group E | Pre-test | 15 | 3.878 | 0.221 | 0.057 |
| | post test | 15 | 3.868 | 0.306 | 0.079 |

Control group: Group E = without any aromatherapy intervention; Experimental groups: Group A = massage with compound essential oils + inhalation; Group B = massage with compound essential oils; Group C = massage with pure base oil; Group D = inhalation with compound essential oils.

**Table 3.** The overall pre- and post-test dependent sample *t*-test checklist for the "physical and mental health" of the elderly in the community.

| Group | Pairwise Variable Difference | | | | | t | df | Significance (Two-Tailed) |
|---|---|---|---|---|---|---|---|---|
| | Average | Standard Deviation | Standard Error of the Mean | Confidence Interval for 95% Difference Number | | | | |
| | | | | Lower Limit | Upper Limit | | | |
| Group A | −0.788 | 0.278 | 0.072 | −0.942 | −0.633 | −1.963 | 14 | 0.000 |
| Group B | −0.254 | 0.250 | 0.065 | −0.393 | −0.116 | −3.933 | 14 | 0.002 |
| Group C | −0.296 | 0.296 | 0.076 | −0.460 | −0.132 | −3.871 | 14 | 0.002 |
| Group D | −0.358 | 0.277 | 0.072 | −0.512 | −0.205 | −5.003 | 14 | 0.000 |
| Group E | 0.010 | 0.230 | 0.059 | −0.118 | 0.137 | 0.164 | 14 | 0.872 |

Control group: Group E = without any aromatherapy intervention; Experimental group: Group A = massage with compound essential oils + inhalation; Group B = massage with compound essential oils; Group C = massage with pure base oil; Group D = inhalation with compound essential oils.

**Table 4.** Summary table of pre- and post-test mean and standard deviation of "stress" for middle-aged and elderly people in the same group.

| Group | Test | Number | Average | Standard Deviation | Standard Error |
|---|---|---|---|---|---|
| group A | Pre-test | 15 | 5.400 | 2.261 | 0.584 |
| | post test | 15 | 2.933 | 2.344 | 0.605 |
| group B | Pre-test | 15 | 2.867 | 1.922 | 0.496 |
| | post test | 15 | 0.867 | 1.246 | 0.322 |
| group C | Pre-test | 15 | 2.267 | 1.668 | 0.431 |
| | post test | 15 | 1.867 | 1.642 | 0.424 |
| group D | Pre-test | 15 | 3.533 | 3.335 | 0.861 |
| | post test | 15 | 1.933 | 2.052 | 0.530 |
| group E | Pre-test | 15 | 3.600 | 2.028 | 0.524 |
| | post test | 15 | 3.067 | 1.870 | 0.483 |

Control group: Group E = without any aromatherapy intervention; Experimental groups: Group A = massage with compound essential oils + inhalation; Group B = massage with compound essential oils; Group C = massage with pure base oil; Group D = inhalation with compound essential oils.

**Table 5.** Sample *t*-test checklist for pre- and post-test dependent samples of "stress" in the same group of middle-aged and elderly people in the community.

| Group | Pairwise Variable Difference | | | | | t | df | Significance (Two-Tailed) |
|---|---|---|---|---|---|---|---|---|
| | Average | Standard Deviation | Standard Error of the Mean | Confidence Interval for 95% Difference Number | | | | |
| | | | | Lower Limit | Upper Limit | | | |
| group A | 2.467 | 1.727 | 0.446 | 1.511 | 3.423 | 5.533 | 14 | 0.000 |
| group B | 2.000 | 1.773 | 0.458 | 1.018 | 2.982 | 4.369 | 14 | 0.001 |
| group C | 0.400 | 1.242 | 0.321 | −0.288 | 1.088 | 1.247 | 14 | 0.233 |
| group D | 1.600 | 1.957 | 0.505 | 0.516 | 2.684 | 3.167 | 14 | 0.007 |
| group E | 0.533 | 1.125 | 0.291 | −0.090 | 1.157 | 1.835 | 14 | 0.088 |

Control group: Group E = without any aromatherapy intervention; Experimental groups: Group A = massage with compound essential oils + inhalation; Group B = massage with compound essential oils; Group C = massage with pure base oil; Group D = inhalation with compound essential oils.

*4.2. Discussion*

4.2.1. Difference Analysis of the Overall Pre- and Post-Test of "Physical and Mental Health" among the Elderly in the Community before and after the Aromatherapy Intervention

From Tables 2 and 3

(1) The average scores for "physical and mental health" in Group A were 3.678 and 4.465, respectively, before and after the test. The t value of the paired sample test was −1.963, $p < 0.05$, which was significant. It showed that there was a difference in the satisfaction of the "physical and mental health" of Group A before and after the test, and that the post-test (M = 4.465) was slightly higher than the pre-test (M = 3.678).

(2) The average score value for "physical and mental health" in Group B was 3.846 and 4.100, before and after the test, respectively. The t value of the paired sample test was −3.933, $p < 0.05$, which was significant. It showed that there was a difference in the satisfaction of Group B's "physical and mental health" before and after the test, and that the post-test (M = 4.100) was slightly higher than the pre-test (M = 3.846).

(3) The average scores for "physical and mental health" in Group C were 3.760 and 4.056, respectively, before and after the test. The t value of the paired sample test was −3.871, $p < 0.05$, which was significant. It showed that there was a difference in the satisfaction level of "physical and mental health" in Group C before and after the test, and that the post-test (M = 4.056) was slightly higher than the pre-test (M = 3.760).

(4) The average scores for "physical and mental health" in Group D were 3.653 and 4.011 in the pre- and post-test, respectively. The t value of the paired sample test was −5.003, $p < 0.05$, which was significant. It showed that there was a difference in the satisfaction level of "physical and mental health" in Group D before and after the test, and that the post-test (M = 4.011) score was slightly higher than the pre-test (M = 3.653).

(5) The average scores for "physical and mental health" in Group E were 3.878 and 3.868, respectively, before and after the test. The t value of the paired sample test was 0.164, $p > 0.05$, which did not reach significance. It showed that there was no difference in the satisfaction of the "physical and mental health" in Group E before and after the test.

Groups A, B, C, and D showed significant differences in the overall "physical and mental health" of middle-aged and elderly people before and after the aromatherapy intervention, but there was no significant difference in Group E. The possible reason was that Group E did not use aromatherapy intervention resulting in no significant difference.

The study focused on the differences and changes between all subjects before and after the experiment. The significant difference in satisfaction for "physical and mental health" before and after the use of aromatherapy indicates that the aromatherapy intervention was beneficial to the physical and mental health of the elderly in the community and can be accepted by the elderly in the community.

4.2.2. Difference Analysis of the Overall Pre- and Post-Test of "Stress" among the Elderly in the Community before and after the Aromatherapy Intervention

From Tables 4 and 5

(1) The average "pressure" of Group A was 5.400 and 2.933, before and after the test, respectively. The t value of a paired sample test was 5.533, $p < 0.05$, reaching a significant level, indicating that the satisfaction of Group A in the pre-test and post-test was different, and that the post-test (M = 2.933) was lower than the pre-test (M = 5.400).

(2) The mean values of "pressure" in Group B were 2.867 and 0.867 in the pre-test and post-test, respectively. The t value of the paired sample test was 4.369, $p < 0.05$, reaching a significant level, which showed that the satisfaction of Group B in the pre-test and post-test was different, and that the post-test (M = 0.867) was lower than the pre-test (M = 2.867).

(3) The mean values of "pressure" in Group C were 2.267 and 1.867 in the pre- and post-test, respectively. The t value of this paired sample test was 1.247, $p > 0.05$, which did not reach a significant level. It showed that there was no significant difference in the satisfaction of Group C before and after the "stress" test.

(4) The mean values of "pressure" in Group D were 3.533 and 1.933 in the pre- and post-test, respectively. The t value of the paired sample test was 3.167, $p < 0.05$, which reached a significant level. It showed that there was a difference in satisfaction for "stress" in the pre-test and post-test in group D, and that the post-test (M = 1.933) was lower than the pre-test (M = 3.533).

(5) The mean values of "pressure" in group E were 3.600 and 3.067 in the pre- and post-test, respectively. The t value of the paired sample test was 1.835, $p > 0.05$, which did not reach a significant level. This result indicated that there was no significant difference in the satisfaction of Group E before and after the "stress" test.

The observations for the experiments "before" and "after" the treatment with aromatherapy, showed that there were also significant differences in the "stress" of middle-aged and elderly people in Groups A, B, and D, but there was no significant difference observed in Groups C and E. The possible reason was that Group C only used base oil and did not use compound essential oils, while Group E did not use aromatherapy intervention resulting in no significant difference

From the results of this study, we can conclude that there was a significant difference in the "stress" level of the elderly in the same groups before and after the aromatherapy intervention.

## 5. Conclusions

*The Use of Aromatherapy Can Improve the "Physical and Mental Health" of the Elderly in the Community and Reduce Their "Stress"*

The significant difference in the satisfaction with "physical and mental health" before and after using aromatherapy indicates that an aromatherapy intervention is beneficial to the physical and mental health of the elderly in the community and can be accepted by the elderly in the community.

Group A, Group B, and Group D showed significant differences in "stress" before and after using aromatherapy, indicating that the aromatherapy intervention was helpful for stress relief among the elderly in the community, and can be used by the elderly in the community. Research has confirmed that the 5% concentration of the essential oil blend, formulated with lavender essential oil, bergamot essential oil, and marjoram essential oil, has a synergistic effect and can effectively improve the physical and mental health of middle-aged and elderly people and reduce stress. Therefore, it is suggested that middle-aged and elderly people can use aromatherapy to improve their physical and mental health and stress.

It is recommended that aromatherapy is incorporated into combination therapy and is used as part of a comprehensive rehabilitation approach. Middle-aged and elderly people should be enabled to use aromatherapy to improve their physical and mental health and stress.

The above experimental results for aromatherapy provide a basis for the home application of aromatherapy for elderly people in the community and for health promotion courses and practical application programs for the elderly.

**Author Contributions:** Conceptualization, M.-H.K. and K.-T.H.; methodology, W.-Y.H.; software, K.-T.H.; validation, M.-H.K. and K.-T.H.; formal analysis, M.-H.K.; investigation, K.-T.H.; resources, K.-T.H.; data curation, M.-H.K.; writing—original draft preparation, M.-H.K.; writing—review and editing, M.-H.K.; visualization, W.-Y.H.; supervision, W.-Y.H.; project administration, K.-T.H.; funding acquisition, K.-T.H. All authors have read and agreed to the published version of the manuscript.

**Funding:** This research received no external funding.

**Institutional Review Board Statement:** This study has been approved by the Human Research Ethics Governance Framework of National Cheng Kung University, Taiwan, and the Human Research Ethics Review Committee. Approval date: 14 November 2019.

**Informed Consent Statement:** Written informed consent has been obtained from the patient(s) to publish this paper.

**Data Availability Statement:** Due to the application for ethical review, the storage and confidentiality planning of the research data is that after the paper data is entered into the computer, it will be stored until July 2020 and then destroyed, and the electronic data will be stored until July 2022. It will be deleted every month and will not be disclosed to the public.

**Conflicts of Interest:** The authors declare no conflict of interest.

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
