# Peer review of "Effects of Aromatherapy on the Physical and Mental Health and Pressure of the Middle-Aged and Elderly in the Community"

_applsci, doi:10.3390/app12104823_

Round 1
Reviewer 1 Report
The attached article of the author's team focuses on the issues of Effects of Aromatherapy on the Physical, Mental Health and Pressure of the Middle-aged and Elderly in the Community. First of all, it is a relatively high-quality processing of the presented information. I consider the processing of theoretical starting points to be very good. These starting points introduce the reader to the issue of Aromatherapy. From my point of view, the elaboration is of high quality and is supplemented by a number of professional studies / articles. From my point of view, the introduction of the article is at a very high quality level and points to the fact that the authors are properly oriented in the given issue and offer a comprehensive view. In the case of the methodology, again, I do not find any serious shortcomings. The authors suitably supplemented the limits of the study, which I would rather include beyond the results, or to the discussion. But I will leave it to the authors' discretion, it does not change the quality of the work, and the inclusion of the study's limits also has its justification in the current location in the text of the article. I have the following reservations about the article:
1. the authors work with human subjects, but in the text I miss the approval or statement of the ethics committee. Is it possible to supplement it? So were the probands informed? Did they sign an informed consent? I realize that these are not invasive procedures, but they are still about working with the human subject. In this case, I would recommend incorporating this ethical perspective.
2. I recommend better summarizing the results and also focusing on this interpretation and drawing conclusions and recommendations for practice.
3. In order to strengthen the originality of the article, I recommend to mention in the paragraph a paragraph on combination therapy, which follows relatively well on the last sentence of the text. We perceive combination therapy as a modern way of comprehensive rehabilitation for seniors. This therapy affects stress, cognitive and motor functions. I therefore recommend mentioning that aromatherapy can be part of comprehensive rehabilitation approaches.
The addition of the general idea of combination therapy on the basis of the submitted sample articles will close the quality and originality of the submitted article and subsequently the given article will be fully prepared for publication from my point of view.
Author Response
1.、the authors work with human subjects, but in the text I miss the approval or statement of the ethics committee. Is it possible to supplement it? So were the probands informed? Did they sign an informed consent? I realize that these are not invasive procedures, but they are still about working with the human subject. In this case, I would recommend incorporating this ethical perspective.
Response 1:(Supplementary to 3.1 Research design)
2.、I recommend better summarizing the results and also focusing on this interpretation and drawing conclusions and recommendations for practice.
Response 2:( Corrected)
3、 In order to strengthen the originality of the article, I recommend to mention in the paragraph a paragraph on combination therapy, which follows relatively well on the last sentence of the text. We perceive combination therapy as a modern way of comprehensive rehabilitation for seniors. This therapy affects stress, cognitive and motor functions. I therefore recommend mentioning that aromatherapy can be part of comprehensive rehabilitation approaches.
Response 3:( Supplemented in 2.3 The application of aromatherapy in combination therapy, the suggestion is also mentioned).

Reviewer 2 Report
The authors conducted a study entitled "Effects of Aromatherapy on the Physical, Mental Health and Pressure of the Middle-aged and Elderly in the Community" based on previously published studies. General comments, unfortunately I cannot recommend the present manuscript for publication, as I have not been able to identify enough scientific advances to justify its publication in this important scientific journal.
Author Response
Thanks for the guidance of experts and scholars, hard work
Reviewer 3 Report
This is a paper investigating the effects of aromatherapy on the physical, mental health and pressure of the middle-aged and elderly in the community. The paper is well-written and of interest for the readers; however, several changes should be made before publishing it.
The abstract section is too long, particularly the introduction section of the abstract. In this section, the authors reported that the research aims to explore the physical and mental health, stress and other issues. However, the main aim is to investigate the effects of aromatherapy. Goals should be clarified.
Study design should be described in the abstract section. How were the patients randomized to the interventions?
The main purpose of the study should be reported at the end of the introduction section.
The subsections 1.3. and 1.4. should not be placed in the introduction.
The "Research objects" section should be replaced into a Material and Methods section.
Research limitations should be stated at the end of the discussion section.
The section called Literature review should be included in the introduction section. If it is an objective of the study, it should be previously reported. If not, it is information to include in the introduction section.
Research methods section should be renamed as "Material and Methods". In this section, the authors should describe the study design, characteristics of the participants, inclusion and exclusion criteria. In a subsection of the material and methods, the authors should include the "Research tools".
In the results section, the authors are reporting data that are presented in Tables 4-1 and 4-2. Data can be described but not repeated in both parts.
Prior to the conclusions section, a discussion should be provided. The conclusions should be brief and open to future studies.
Author Response
1、The abstract section is too long, particularly the introduction section of the abstract. In this section, the authors reported that the research aims to explore the physical and mental health, stress and other issues. However, the main aim is to investigate the effects of aromatherapy. Goals should be clarified.
Response 1: (The abstract is simplified and revised, and the target content is supplemented)
2、Study design should be described in the abstract section. How were the patients randomized to the interventions? .
Response2:( Described as intentional sampling)
3、The main purpose of the study should be reported at the end of the introduction section The subsections 1.3. and 1.4. should not be placed in the introduction.
Response 3:(removed, corrected)
4、The "Research objects" section should be replaced into a Material and Methods section.
Response 4:(processed)
5、Research limitations should be stated at the end of the discussion section.
Response 5:( processed)
6、The section called Literature review should be included in the introduction section. If it is an objective of the study, it should be previously reported. If not, it is information to include in the introduction section.Research methods section should be renamed as "Material and Methods". In this section, the authors should describe the study design, characteristics of the participants, inclusion and exclusion criteria.
Response6:( This part 3.2 The research object has been explained)
7、In a subsection of the material and methods, the authors should include the "Research tools".
Response 7:(This section 3.4 Research tools has been explained)
8、In the results section, the authors are reporting data that are presented in Tables 4-1 and 4-2. Data can be described but not repeated in both parts. Response 8:(removed, corrected)
9、Prior to the conclusions section, a discussion should be provided. The conclusions should be brief and open to future studies.
Response 9:(Additional discussion)

Round 2
Reviewer 2 Report
Authors performed important reviews, I recommend MS for publication
Author Response
Thank you for your expert review

Reviewer 3 Report
The structure of the paper can be improved.
The introduction is too long and the research methods do not need to include limitations. This last part (limitations) should be included at the end of the manuscript.
The results and discussion section should be separated.
Author Response
1、The structure of the paper can be improved.
Response 1:(The first round has provided the round review report)
2、The introduction is too long and the research methods do not need to include limitations. This last part (limitations) should be included at the end of the manuscript.
Response 2: (3.3 Research limitations removed)
3、The results and discussion section should be separated
Response 3:( First round processed)
